# Characterization of SARS-CoV-2 Evasion: Interferon Pathway and Therapeutic Options

**DOI:** 10.3390/v14061247

**Published:** 2022-06-08

**Authors:** Mariem Znaidia, Caroline Demeret, Sylvie van der Werf, Anastassia V. Komarova

**Affiliations:** Génétique Moléculaire des Virus à ARN Unit, Department of Virology, Institut Pasteur, Université Paris Cité, CNRS UMR3569, 75015 Paris, France; mariem.znaidia@pasteur.fr (M.Z.); caroline.demeret@pasteur.fr (C.D.); sylvie.van-der-werf@pasteur.fr (S.v.d.W.)

**Keywords:** SARS-CoV-2, innate immunity, virus–host interactions, antagonism, interferon, therapy

## Abstract

Severe acute respiratory syndrome coronavirus-2 (SARS-CoV-2) is responsible for the current COVID-19 pandemic. SARS-CoV-2 is characterized by an important capacity to circumvent the innate immune response. The early interferon (IFN) response is necessary to establish a robust antiviral state. However, this response is weak and delayed in COVID-19 patients, along with massive pro-inflammatory cytokine production. This dysregulated innate immune response contributes to pathogenicity and in some individuals leads to a critical state. Characterizing the interplay between viral factors and host innate immunity is crucial to better understand how to manage the disease. Moreover, the constant emergence of new SARS-CoV-2 variants challenges the efficacy of existing vaccines. Thus, to control this virus and readjust the antiviral therapy currently used to treat COVID-19, studies should constantly be re-evaluated to further decipher the mechanisms leading to SARS-CoV-2 pathogenesis. Regarding the role of the IFN response in SARS-CoV-2 infection, in this review we summarize the mechanisms by which SARS-CoV-2 evades innate immune recognition. More specifically, we explain how this virus inhibits IFN signaling pathways (IFN-I/IFN-III) and controls interferon-stimulated gene (ISG) expression. We also discuss the development and use of IFNs and potential drugs controlling the innate immune response to SARS-CoV-2, helping to clear the infection.

## 1. Introduction

Severe acute respiratory syndrome coronavirus-2 (SARS-CoV-2) is responsible for the current COVID-19 pandemic that emerged in Wuhan, Hubei, China, in December 2019 [1]. SARS-CoV-2 belongs to the Coronaviridae family. It is an enveloped positive-sense single-stranded RNA virus with a genome of 29,903 nucleotides that replicates entirely in the cytoplasm. The SARS-CoV-2 genome is composed of thirteen recognized open reading frames (ORFs) [2,3]. Upon the release of the SARS-CoV-2 genome into the cytoplasm, ORF1ab is translated to polyproteins pp1a and pp1ab. These polyproteins are then cleaved by viral chymotrypsin-like (3CLpro or NSP5) and papain-like (PLpro or NSP3) proteases to give 16 nonstructural proteins (NSP) that encode the RNA-dependent RNA polymerase (RdRp), helicase, and other components required for virus replication. Similar to other coronaviruses, SARS-CoV-2 replication involves the synthesis by RdRp of positive- and negative-sense full-length genomes. During its replication, SARS-CoV-2 also produces a nested set of subgenomic RNAs (sgRNAs). SARS-CoV-2 sgRNAs encode four structural proteins (S, spike; E, envelope; M, membrane; and N, nucleocapsid) and several accessory factors (ORF3a, ORF3b, ORF6, ORF7a, ORF7b, ORF8, and ORF9b) [1,4,5,6,7].

Innate immunity is the first line of defense against pathogens. It plays a critical role in controlling viral infections by mounting a rapid antiviral state and by mediating the adaptive immune response [8]. Interferons (IFNs) are transcriptionally regulated cytokines that play a central role in the innate antiviral immune response and orchestrate adaptive immunity. In some cases, this immune response could participate in the pathology. Indeed, SARS-CoV-2 infection is characterized by a dysregulated innate immune response, which contributes to the pathogenicity, leading in some individuals to a critical state [9,10,11]. An early IFN response is needed to establish a robust antiviral state, but this response is weak and delayed in severe COVID-19 patients, along with massive pro-inflammatory cytokine production [12,13,14].

The first step of innate immunity is the recognition of pathogen associated molecular patterns (PAMPs) by pattern-recognition receptors (PRRs) that initiates signaling cascades leading to the production of IFN, pro-inflammatory cytokines, and chemokines [15]. Among PRRs, Toll-like receptor 3 (TLR3), TLR7, TLR8, RIG-I-like receptors (retinoic acid-inducible gene I (RIG-I)), and melanoma differentiation-associated protein 5 (MDA5) have been described to recognize non-self RNA molecules [16,17]. RIG-I is specialized in the recognition of the 5′-triphosphate end of relatively short double-stranded (ds)RNAs, while MDA5 recognizes long dsRNAs [18]. SARS-CoV-2 RNAs have been described to be recognized by RIG-I and MDA5 [19,20], with MDA5 being the main cytoplasmic PRR capable of detecting SARS-CoV-2 [21,22]. RIG-I and MDA5 are composed of an RNA helicase domain and a C-terminal domain (CTD) involved in viral RNA recognition. The N-terminal caspase activation and recruitment domains (CARDs) of both RIG-I and MDA5 interact with adaptor proteins of the mitochondrial-associated viral signaling proteins (MAVS) to trigger IFN regulatory factor 3 (IRF3) and IRF7 phosphorylation and dimerization, leading to their translocation to the nucleus and the transcription of IFN-I and IFN-III. The NF-κB signaling cascade is also activated through the so-called canonical pathway by a set of receptors that includes TLRs and RLRs, triggering the expression of pro-inflammatory cytokines and chemokines, such as interleukin 6 (IL-6), IL-1B, tumor necrosis factor (TNF)- alpha, and chemokine ligand 2 (CCL2) [23,24]. IFN-I and IFN-III play a central role in antiviral immunity. IFN-α and β belong to IFN-I, while IFN-λ (IFN-λ: λ1, λ2, λ3, and λ4) belongs to IFN-III. Both IFN-I and IFN-III are secreted by infected cells and are involved in inhibiting viral infections. However, their action involves distinct receptors. IFN-α receptor 1 (IFNAR1) and IFNAR2, the two chains forming the heterodimer at the cell membrane surface, are indispensable to IFN-I binding and subsequent signaling. Contrary to IFN-I receptors, which are expressed by most cells, the IFN-λ: λ1, λ2, λ3, and λ4 (IFN-LR1) chain of the IFN-III receptor (IFN-LR1/ IL-10R2 heterodimer) is expressed by a restricted subset of cells, including epithelial cells [25]. IFN-I and IFN-III share an overlapping pathway of activation and functions, but IFN-I has been described to exert a more potent and rapid response, while IFN-III expression is more sustained and restricted to epithelial barriers. Both are secreted by infected cells, leading to an autocrine and paracrine stimulation through the JAK/STAT signaling pathway involving signal transducers and activators of transcription (STAT1 and STAT2), protein phosphorylation by Janus kinase 1 (JAK 1) and tyrosine kinase 2 (TYK2), and dimerization to interact with IRF9. Together, STAT1, STAT2, and IRF9 form transcription factor IFN-stimulated gene factor 3 (ISGF3), which translocates to the nucleus and binds to IFN-stimulated response elements (ISREs) in interferon-stimulated gene (ISG) promoters, resulting in their activation [26,27]. These ISGs play a crucial role in repressing viral replication via various mechanisms. They prevent viral entry into the cell and viral trafficking into the nucleus, inhibit viral replication along with viral transcription/translation, and can also degrade viral nucleic acids and block viral particle assembling [28,29].

SARS-CoV-2 has developed multiple strategies to impede this innate immune response and achieve efficient replication. The knowledge of the mechanisms by which SARS-CoV-2 interferes with innate immunity is crucial to apprehending this newly emerged virus and developing vaccines and therapies. In this review, we describe recent advances in the understanding of the circumvention of the innate immunity by SARS-CoV-2 (summarized in Table 1 and represented in Figure 1) and explore some therapeutic options that were developed and assessed for COVID-19 (Table 2).

## 2. Evasion of Innate Immunity by SARS-CoV-2

### 2.1. Evasion of Sensing by Host Innate Immune Receptors

Coronaviruses, including SARS-CoV-2, shield their dsRNA replication intermediates in double-membrane vesicles (DMVs) mediated by NSP3, NSP4, and NSP6, avoiding dsRNA sensing by RIG-I, MDA5, and TLR3 [30,31]. Moreover, viral RNA capping is a critical step to prevent PRR sensing. It is mediated by the RNA triphosphatase activity of NSP13, the RdRp NSP12, the NSP14 N7-MTase activity, NSP10, and NSP16 2′-O-MTase activity [32]. Since RIG-I recognizes RNA 5′-triphosphate, by harboring a ribose 2′-O-methylation 5′-cap structure, SARS-CoV-2 evades innate immune recognition [33,34]. The nucleocapsid (N) has been described to inhibit SARS-CoV-2 RNA recognition by RIG-I. The first molecular mechanism has been proposed where RIG-I inhibition is mediated by a direct interaction through its DExD/H helicase domain with the N protein [35]. This domain has an ATPase activity playing a crucial role in RNA binding, suggesting N directly blocks the RNA recognition.

The activation of PRRs can also be inhibited. ISG15, a ubiquitin-like protein that can be covalently conjugated to lysine residues (ISGylation), plays an essential role in MDA5 activation. NSP3 cleaves ISG15 [36] and consequently antagonizes ISG15-dependent MDA5 activation, evading immune recognition [37]. Additionally, ISG15 acts as an antiviral factor, modifying host and viral proteins, which interferes with viral assembly or function. The other protease, NSP5, further mediates the inhibition of PRR functioning. RIG-I is cleaved by NSP5 at the last 10 N-terminal amino acids, blocking its ability to signal through MAVS [38]. In addition, other studies have described the suppression of IFN-β production in the consequence of TRIM25’s interaction with the N protein [39,40]. The ubiquitination of the RIG-I CARD domain by TRIM25 leads to the activation of RIG-I, leading to downstream signaling. By this N-TRIM25 interplay, the signaling cascade is blocked [40,41].

### 2.2. Inhibition of Innate Immune Receptor Signaling and IFN Production

Strategies have been mounted by SARS-CoV-2 to inhibit the activation of PRRs downstream of the PAMPs sensing step. IFN-I and IFN-III responses are fundamental defenses of the antiviral innate immunity in the clearance of infection. To establish a successful infection of host cells, SARS-CoV-2 has developed various strategies to antagonize different stages of the signaling leading to IFN production. Particularly, IFN-β production is inhibited by NSP1, NSP3, NSP5, NSP12, NSP13, NSP14, NSP15, ORF3a, ORF3b, ORF6, ORF7a, ORF7b, ORF8, ORF9b, N, and M, as reported by several studies [42,43,44,45,46]. These proteins have been shown to reduce RIG-I-mediated IFN-β promoter activities, suggesting they can suppress RLR-mediated signaling [19,42,43,44]. This is mediated by many molecular strategies (Figure 1 and Table 1). Among them, are the inhibition of IRF3 phosphorylation and its nuclear translocation, which are pivotal steps required for the activation of IFN-I transcription [47,48]. NSP1 [49], NSP12 [50], NSP13, NSP14, NSP15, ORF3b [51], and ORF6 [42] have been reported to inhibit the nuclear translocation of IRF3. For example, the NSP12 inhibition of IRF3 nuclear translocation impairs downstream signaling of type-I IFN, as ISRE and ISG56 promoters have been described to be inhibited in this way [50]. However, in a study by Li et al., no IRF3 nuclear translocation inhibition was shown when NSP12 was overexpressed [52]. This study demonstrated that NSP12 could reduce IFN-β luciferase promoter activity via a yet-to-be-determined mechanism but cannot inhibit the IFN-β production induced by Sendai virus infection (well-known RLR agonist) or another stimulus as well as the downstream signaling. ORF6 has been described to bind importin karyopherin α 2 (KPNA2). Since KPNA2 is an importin involved in importing IRF3, the inhibition of IRF3 translocation in the presence of ORF6 has been suggested to be due to its interaction with KPNA2 [44]. IRF3 activation and subsequent IFN production depend on the assembly of the multiprotein complex containing RIG-I/MDA-5, MAVS, TNF receptor-associated factor 3 (TRAF3), and TANK-binding kinase 1 (TBK1). TBK1 is a protein kinase that is involved in IFN-I transcription by phosphorylating IRF3 [53]. The M protein prevents the formation of this complex [54] by association with TBK1 and its degradation via the ubiquitin pathway, thereby inhibiting IRF3 phosphorylation and IFN production [55,56]. Besides, ORF7a reduces IRF3 nuclear translocation by destabilizing TBK1 [19]. NSP13 also inhibits TBK1 phosphorylation, resulting in decreased IRF3 activation and IFN-β production [57]. NSP6 and NSP15 bind TBK1 to decrease IRF3 phosphorylation as well [44]. ORF9b interacts with the viral RNA sensors RIG-I, MDA-5, and the downstream signaling factors that include, for example, MAVS, and TBK1, leading to TBK1 phosphorylation inhibition, consequently inhibiting IRF3 phosphorylation, its nuclear translocation, and types I and III IFN transcription [58]. Furthermore, ORF9b has been described to associate with TOM70, an adaptor protein of the MAVS complex at the mitochondrial membrane, inhibiting the triggering signal leading to IFN production [59]. Another antiviral role of ORF9b is to interact with NF-κB essential modulator (NEMO) to block its K63-linked polyubiquitination. This ubiquitylation is necessary to activate the IKKα/β kinase phosphorylation function of NEMO, leading to IKKα ubiquitylation and degradation and the subsequent translocation of NF-κB subunit p65 into the nucleus to activate the transcription of pro-inflammatory cytokines [60]. Furthermore, the translocation of p65 is repressed because of NSP9’s interaction with the nucleoporin NUP62 [61]. By its ability to alter nuclear pore complex (NPC) composition, NSP9 is consequently capable of blocking the antiviral immune response. Furthermore, NSP5 inhibits the pro-inflammatory NF-κB pathway [62]. Moreover, NSP5 also prevents the nuclear translocation of IRF3 without direct cleavage [63], while other studies have demonstrated an IRF3 degradation promoted by NSP5 [62,64]. Additionally, NSP5 promotes the ubiquitination and proteasome-mediated degradation of MAVS [38]. Besides, the direct cleavage of IRF3 by NSP3 has been demonstrated by Moustaqil et al. [62]. This huge redundancy in antagonizing IFN production highlights the importance of SARS-CoV-2 to shutting down the IFN response to the success of its infection.

### 2.3. Inhibition of IFN Signaling and ISG Expression

SARS-CoV-2 has also developed many strategies to antagonize the IFN response, inhibiting IFN-α/β receptor signaling and ISG expression. The very early step of IFN signaling is targeted by SARS-CoV-2. NSP14 mediates IFNAR1 lysosomal degradation, blocking the activation of the transcription factors STAT1 and STAT2 [65]. Additionally, the phosphorylation of STAT1 and STAT2 is counteracted by several viral proteins. Xia et al. have described the capacity of NSP1, NSP6, NSP13, ORF3a, M, ORF6, ORF7a, and ORF7b to suppress >40% of ISRE promoter activity by inhibiting STAT1 phosphorylation for NSP1, NSP6, NSP13, ORF3a, ORF7b, M, ORF6, and ORF7a and STAT2 phosphorylation for NSP1, ORF6, ORF3a, and M [44]. ORF6 impairs IFN production and has also been described to block STAT1/2 nuclear translocation, a crucial step to activate the transcription of ISGs [66]. Indeed, ORF6 interacts with the nucleopore complex Nup98 via its C-terminus, thus blocking mRNA export [67]. This leads to the impairment of IFN mRNA nuclear export and also transcription factor nuclear import. The ORF6 nucleopore complex interaction has been proposed to clog the nuclear pore [68], explaining the inhibition of STAT1 and IRF3 nuclear translocations [43,44]. The N protein has been demonstrated to competitively bind to STAT1/STAT2, thus interfering with the interactions of STAT1 with JAK1 and STAT2 with TYK2, respectively. This inhibits their phosphorylation and subsequent ISGs production [69]. Moreover, ORF7a hijacks the host ubiquitin system. The host ubiquitin system is usurped to form K63-linked ubiquitin chains, enhancing, in this way, its ability to inhibit STAT2 phosphorylation and, hence, blocking ISGF3 complex translocation to the nucleus to activate ISG transcription [70].

Thus, the IFN signaling pathway is blocked at different levels by several SARS-CoV-2 proteins, ensuring an advantage for viral replication. The ISGs produced are also targeted. Viral proteases have been found to cleave IFN-stimulated antiviral proteins as soon as they are produced. For instance, NSP3 cleaves ISG15 [36].

### 2.4. Inhibition of Host Protein Production by Targeting Post-Transcriptional and Translational Steps

NSP16 interferes with mRNA production. The critical step of mRNA splicing is mediated by the spliceosome, a complex of proteins and non-coding RNAs. NSP16 inhibits the splicing of pre-mRNA by binding to U1 and U2, which are two small nuclear RNAs involved in this complex. The ISG expression is then repressed [71]. mRNA export to the cytoplasm has been described to be blocked by NSP1 [72]. Mechanistically, NSP1 interacts with the host mRNA export receptor heterodimer NXF1-NXT1 [73]. This receptor is responsible for the nuclear export of cellular mRNAs by binding to mRNA and docking at the NPC. NSP1 binds to NXF1, reducing the interaction with the NPC and preventing mRNA nuclear transport. Moreover, the NSP1 C-terminal domain binds to the small ribosomal subunit (40S), more specifically to the 18S rRNA, disrupting 40S mRNA scanning and preventing translation initiation, then blocking host protein production [71]. This impacts IFN production without altering viral mRNA translation. Cryo-electron microscopy analyses of the molecular interaction between NSP1 and the 40S ribosome subunit have revealed that NSP1 is bound to the mRNA entry channel overlapping the mRNA path [74,75,76,77]. However, how viral mRNA containing the leader sequence passes through the translation shut off triggered by NSP1 is not fully understood. Banerjee et al. have observed that the common SARS-CoV-2 5′-leader sequence precisely positions viral mRNAs relative to the NSP1-40S complex, enabling translation [71]. Mendez et al. demonstrated that the alteration of NSP1-mRNA-40S complex by mutating residues within the N-terminal domain of NSP1 makes a weaker association with the ribosome and mRNA, abrogating the translational escape of viral leader-containing mRNAs. They suggested that these NSP1 residues are important to allow the viral leader-sequence-triggered conformational changes to NSP1 that support viral mRNA translation [78]. NSP8 and NSP9 disrupt protein trafficking, leading to the degradation of newly translated proteins. The signal recognition particle (SRP) is a complex that binds the 80S ribosome and identifies co-translationally hydrophobic signal peptides present in secreted and membrane proteins, leading to their translocation to the endoplasmic reticulum and their proper folding and expression at the membrane. NSP8 and NSP9 bind to 7SL RNA, a non-coding RNA that is a part of the SRP complex, leading to the inhibition of signal peptide recognition. This disruption of protein trafficking suppresses the IFN response, as described by Banerjee et al. [71]. Thus, SARS-CoV-2 developed several strategies to alter general cellular processes involved in post-transcriptional and translational processes.

## 3. Therapeutic Options

SARS-CoV-2 employs many strategies to impede the IFN response, as described above, resulting in an imbalance in the innate responses, with excessive cytokine production [12,13,14]. A weak and delayed IFN response along with an exacerbated inflammatory response has been associated with COVID-19 pathogenesis [9,10,11]. Patients with no IFN-α production present poorer outcomes and show a longer intensive care unit stays [79]. Clinical data support the implication of deficient IFN-I responses in the progression of COVID-19 toward a more severe manifestation [80]. Moreover, studies have shown that COVID-19 severity is enriched in patients with genetic defects in TLR3, IRF7, and TLR7, leading to a deficient induction of IFN-I [81,82,83]. Besides, autoantibodies targeting IFN-I were found in 10% of patients with severe COVID-19, while they were absent in asymptomatic or mildly symptomatic patients. These antibodies were then identified as a risk factor for life-threatening COVID-19 [84,85,86,87] and have been described to increase in patients over 60 years old and cause about 20% of all fatal COVID-19 cases [88]. Many studies have exposed the capacity of both IFN-I and IFN-III to inhibit SARS-CoV-2 in vitro or in vivo [13,89,90,91,92,93]. These observations highlight the crucial role played by the IFN response in controlling SARS-CoV-2 infection and subsequent clinical implications. Therefore, IFN therapy was naturally considered to support innate immunity and control the SARS-CoV-2 infection. Below, we discuss studies that have been performed to test various IFN therapies, which are also summarized in Table 2.

### 3.1. IFN-α

IFN-α has already been used in the viral infection context. Indeed, pegylated IFN-α2 (PEG-IFN-α2, a long-acting form of IFN-α2b) was used as the standard in the treatment of chronic hepatitis C virus infection until it was replaced by antivirals and is still a therapeutic option in mild to moderate chronic hepatitis B [94,95].

In the context of COVID-19, some studies testing IFN-α efficacy have been conducted. Inhaled IFN-α2b administration significantly reduced the duration of SARS-CoV-2 detection in the upper respiratory tract by a reverse transcription-quantitative PCR (RT-qPCR) test and blood levels of inflammatory markers such as IL-6 and C-reactive protein, as reported in an exploratory study by Zhou et al. [96]. A reduction in viral load and clinical status improvement in moderate COVID-19 cases after subcutaneous administration of pegylated IFN-α2b was also reported in a phase II clinical trial and a phase III clinical trial [97,98], but no difference was observed for inflammatory biomarkers or in the duration of hospitalization. Furthermore, patients who have been administrated subcutaneous IFN-α2b within 72 h following admission had shorter hospital stays compared with late administration, as presented by B.Wang et al. [99]. However, the studies described above are limited by the small number of subjects enrolled (40 in [97] and 41 in [99]). Additionally, in the phase III study from N.Wang et al., a reduction in in-hospital mortality was observed in patients who were given IFN-α2b by inhalation using an air compressor machine (nebulized delivery) within 5 days of admission. However, late initiation of this therapy was associated with increased mortality [100]. Similarly, in the study reported by Yu et al., early administration of IFN-α2b aerosol was associated with improved clinical outcomes (such as a lower risk of disease progression and a shorter hospitalization time), but delayed IFN-α2b intervention was associated with increased probabilities of adverse events [101].

At the molecular level, the SARS-CoV-2 receptor ACE2 was reported as an ISG upregulated by IFN-α [102], raising concerns about potential pro-viral effects of ACE2 induction upon IFN treatment. Later, molecular characteristic studies demonstrated that IFNs only induced the expression of a truncated product of ACE2 that is not able to bind the SARS-CoV-2 spike protein [103,104,105].

Thus, the timing of IFN-α administration seems to play a crucial role in its efficacy, and early administration may lower the risk of worse outcomes. Because of the exclusion of patients at risk in these studies, an understanding of the association between age, comorbidities, the stage of the disease, timing, and the IFN-α treatment is still needed. Furthermore, most clinical studies have been focused on nebulized IFN-α, but this form is not yet approved by the United States Food and Drug Administration (FDA), and its use is not recommended and is restricted in clinical studies by the World Health Organization (WHO). Overall, IFN-α treatment remains controversial, even though the outcomes of early administration of IFN-α are encouraging.

### 3.2. IFN-β

Since in vitro studies have reported that SARS-CoV-2 is more sensitive to IFN-β than IFN-α [90], and autoantibodies against IFN-α but not against IFN-β have been found in patients with severe disease [84], IFN-β seemed to be a more appropriate therapy than IFN-α.

Different strategies are employed by SARS-CoV-2 proteins to specifically antagonize IFN-β production, as described in Part 2 of this review. The direct delivery of IFN-β could, in principle, overcome these evasion strategies. Indeed, IFN-β binds IFNAR and initiates the JAK-STAT signaling pathway, thereby inducing ISG expression. IFN-β displays anti-inflammatory properties by enhancing IL-10 expression, and it has been reported to upregulate CD73 on endothelial cells, preventing vascular leakage [106] and inhibiting leukocyte recruitment [107]. Besides, a study carried out before the COVID-19 pandemic reported that IFN-β injection improves acute respiratory distress syndrome that occurs in the presence of predisposing factors [108].

Despite all these experimental pieces of evidence, clinical trial results for IFN-β are controversial. Systemic or subcutaneous deliveries of IFN-β have been reported to be efficient in shortening the duration of viral shedding and hospital stay [109], reducing mortality [110,111] when combined with other antiviral drugs but only in the early stage of the disease in mild to moderate cases. However, the SOLIDARITY randomized trial reported that 2000 COVID-19 hospitalized patients who received subcutaneous IFN-β1a or IFN-β1a together with the antivirals lopinavir-ritonavir experienced little or no effect on clinical improvement [112]. In fact, the subcutaneous delivery of IFN-β1a to severe COVID-19 patients failed to reduce mortality according to the WHO SOLIDARITY trial outcomes [112] and as reported by Kalil and colleagues [113]. Worse outcomes after treatment with IFN-β1a have also been reported, suggesting the potential harm of using IFN-β in patients with severe disease, such as those on high-flow oxygen, noninvasive ventilation, or mechanical ventilation [113]. Thus, the COVID-19 Treatment Guidelines Panel does not recommend the use of systemic IFN-β for the treatment of hospitalized patients with COVID-19 [114]. The administration of IFN-β directly in the respiratory tract may result in a robust local anti-viral immune response, avoiding the inconvenience of IFN-β systemic exposure. Inhaled IFN-β was examined in a randomized double-blind placebo-controlled phase II trial [115]. Indeed, patients treated with nebulized IFN-β1a showed significantly greater odds of clinical improvement across the WHO Ordinal Scale for Clinical Improvement than those who received a placebo [115]. Still, in a study performed on adult patients with moderate COVID-19, the administration of IFN-β1b by vapor inhalation combined with an antiviral Favipiravir failed to provide benefits [116].

The combination with other treatments in many IFN-β trials makes it difficult to derive a clear conclusion about the performance of IFN-β. Additionally, the size of the sample and the stage of the disease when patients are recruited have to be considered when analyzing all these clinical studies. More data are needed to settle if inhaled IFN-β could be used to treat COVID-19. However, it has to be noted that nebulized IFN-β is not yet approved in the USA and Europe, and the pharmacokinetics and pharmacodynamics of this mode of administration are not fully known.

### 3.3. IFN-λ

IFN-I is able to provide a systemic response along with a local one, whereas IFN-III delivers a protection restricted to epithelial surfaces. Especially in the context of influenza virus infection, type-III IFNs appear to be essential in controlling viral infection in the upper respiratory tract, while IFN-I and IFN-III dispense functional redundancy in the lower respiratory tract [117]. IFN-I triggers systemic and local inflammation by activating and recruiting immune cells to the site of infection. In contrast, IFN-III has been shown to limit viral activities at the local level without being associated with inflammation [118,119,120], conversely restricting the neutrophil inflammatory functions then limiting tissue damage [120,121,122]. In the context of SARS-CoV-2 infection, in vitro and in vivo studies have demonstrated antiviral IFN-III activities in epithelial cells without excessive inflammation [89,91,92,123,124]. Furthermore, the protein expression of IFN-λ2/λ3 were reduced in nasopharyngeal swab samples from COVID-19-positive individuals [125]. Additionally, studies have demonstrated a downregulation of systemic IFN-λ1/λ2 in patients with severe COVID-19, suggesting that a reduced level of IFN-III may impact the severity of the disease [126,127,128].

The abovementioned studies highlight the potential benefits of using IFN-λ2 as an antiviral treatment for SARS-CoV-2. IFN-λ has been proposed to help clear the infection and minimize the severity of COVID-19 [129]. Despite of all these promising benefits, as of today only two clinical trials have been finalized [130,131], while few others are still recruiting. In a randomized placebo-controlled trial reported by Jagannathan et al. on 120 patients with mild to moderate COVID-19, 60 received subcutaneous doses of Peginterferon IFN-λ1a within 72 h of diagnosis. No difference has been observed between the Peginterferon IFN-λ1a and placebo groups in SARS-CoV-2 viral shedding or in the improvement of symptoms [131]. In contrast, the study by Feld et al. described the benefits of subcutaneous Peginterferon IFN-λ in treating mild to moderate cases of COVID-19 who were within 7 days of symptom onset or the first positive test if asymptomatic [130]. Further studies are needed to validate the efficacy of IFN-III treatment of COVID-19.

It should be pointed out that in the context of influenza infection IFN-λ has been reported to contribute to bacterial superinfections [132,133]. Since critically ill COVID-19 patients suffer from bacterial superinfections, caution should be exercised in using IFN-λ as a COVID-19 treatment. Additionally, IFN-λ has not been approved by the FDA for any use. These facts could explain the precautionary approach to limit the use of IFN-λ to clinical trials.

## 4. Conclusions and Perspective

COVID-19 has been characterized by a dysregulation of the immune response that can lead to severe acute respiratory distress syndrome in some people. Multiple strategies have been evolved by SARS-CoV-2 structural, nonstructural, and accessory proteins to counteract viral sensing, IFN production, and IFN signaling pathways. This redundancy in evading innate immunity characterizes SARS-CoV-2 infection (Table 1) and underlines the adaptation to the human host. The inhibition of the IFN response by SARS-CoV-2 viral factors, especially NSPs that are expressed early upon virus cell entry, promotes viral replication and participates in the delay in the IFN response observed in COVID-19 patients.

The direct use of IFN therapy in the context of COVID-19 has been expected to resolve a weak early innate immune reaction and restore a strong and well-timed IFN response capable of inhibiting the virus replication before the immunopathology takes place. However, the IFN response is a fine-tuned process that is not fully understood, and clinical trials reveal disparate results depending on time of treatment initiation and severity state. These contradictory results of IFN therapy’s benefits in handling COVID-19 underline that two central factors should be considered: the time from the beginning of the treatment and the route of administration. Some in vitro and in vivo studies have also described differences in efficacy upon IFN-I pretreatment and post-infection treatment, underlying the interest in using models to calibrate IFN therapies [134,135,136]. Besides, a study showed that high IFN-III expression at the protein level in the upper respiratory tract is associated with a mild disease and, hence, could be protective [137], while another study demonstrated an increase at the transcriptional level for IFN-I/IFN-III mRNA but a decrease of the corresponding proteins [125]. Along the same line, single-cell RNA sequencing on nasopharyngeal swabs of COVID-19 patients has revealed a robust IFN-I-specific gene signature in patients with mild to moderate disease in contrast to an impaired one in severe cases [138]. Conversely, some studies have reported a potent IFN response in critical cases. Elevated IFN-I and III in the lower airways at both the transcriptional and protein levels have been reported in patients with severe COVID-19 [137]. Moreover, a sustained production of IFN-α in the blood has been reported in severe COVID-19 cases in a longitudinal study [139], along with an IFN-I transcriptome signature co-existing with the inflammatory one [140]. Thus, IFNs could play a direct role in exacerbating the inflammatory response to SARS-CoV-2 infection. One hypothesis that could explain the opposite roles played by IFN-I/III during COVID-19 pathogenesis is that the IFN response is blocked by the SARS-CoV-2 counteracting strategy in infected epithelial cells, leading to a massive replication and the subsequent high viral load observed. Then, the produced viral particles could stimulate innate immune cells (monocytes, dendritic cells, and macrophages), which produce substantial amounts of IFNs. Thus, the side-effect observed by late IFN administration could be explained by a potential contribution to the cytokine storm, emphasizing the inflammatory state. Furthermore, the ambivalent properties that depend on the anatomical location of various IFN-producing cells and viral tropism to the upper and low respiratory tracts have to be taken into consideration. This underlines the complex role of IFNs in COVID-19 pathogenesis, and further investigations are needed to decipher their exact implication in the immunopathology and to state the appropriate conditions for the therapy.

The understanding of the mechanisms underlying the host and virus interactions is crucial to the development of drugs and vaccines. In particular, apprehending the process of innate immune response establishment and SARS-CoV-2 evasion could help the development of drugs such as immunomodulators or antiviral drugs that directly target viral immune counteractors. The inhibition of those viral components may attenuate pathogenicity by increasing the naturally early host antiviral response. However, a particularity of SARS-CoV-2 among other coronaviruses is to entertain high mutation capacities [141] that give rise to variants, thus challenging drug development and vaccine efficacy. Targeting conserved proteins is needed because they are less prone to undergo mutations. Conducting a computational screening of FDA-approved drugs against such conserved proteins makes a rapid and efficient method to identify effective treatments that can be tested in vitro and in vivo. In this regard, NSP1 appears to be an attractive target since it is a very conserved protein [142] playing a central role in antagonizing the innate immune response, as described in this review [46,57,73,76]. Montelukast sodium hydrate (which is an FDA-approved drug that stably binds to NSP1) has shown antiviral activity against SARS-CoV-2, with reduced viral replication in cell lines in vitro [143]. GRL0617, another FDA-approved drug, which is a non-covalent inhibitor of SARS-CoV-2’s NSP3 [144], displays promising issues on SARS-CoV-2′s NSP3 [145]. GRL0617 has been demonstrated to block the binding of SARS-CoV-2 NSP3 to ISG15 [36], inhibiting its deubiquitinase activity [146]. These examples underline the benefits of investigating the area of SARS-CoV-2’s conserved proteins that are implicated in innate immunity evasion.

Currently, few antiviral drugs and immunomodulators are approved by the WHO *Therapeutics and COVID-19: living guideline in the treatment of COVID-19* [114]. Among them, remdesivir, an analog of adenine inhibiting RdRp was first recommended to treat severe cases and has been described to show a higher efficacy early in the course of the disease, during the viral replication phase of COVID-19, and to not be so useful during the immunopathology phase [114]. In contrast, immunomodulators were shown to be effective to harness inflammation. IL-6 blockers such as the JAK1/JAK2 inhibitor Baricitinib or the monoclonal antibody Tocilizumab, which antagonizes the membrane-bound and soluble forms of the IL-6 receptor, have been recommended to treat severe or critical COVID-19 patients [114]. Nevertheless, ideal drugs for the treatment of COVID-19 are yet to be discovered. Regarding the emergence of several variants of concern since the beginning of the COVID-19 pandemic, challenging the efficacy of currently developed vaccines, therapies targeting conserved viral proteins and adjusting the effective innate immune response to SARS-CoV-2 should be priorities.

## Figures and Tables

**Figure 1 viruses-14-01247-f001:**
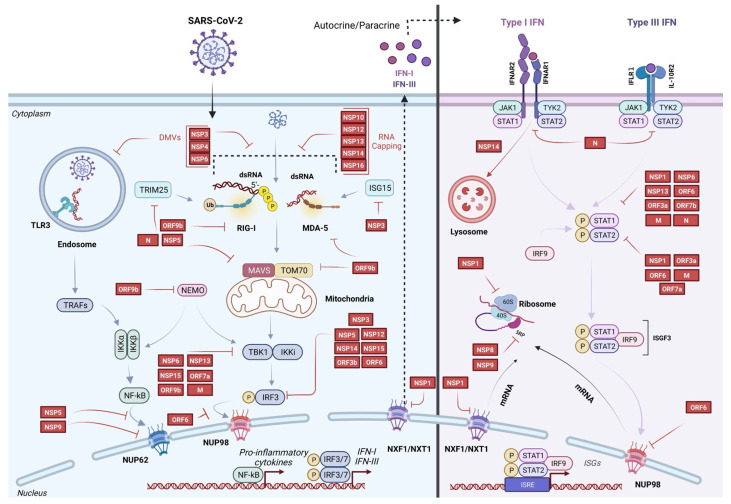
Model describing the pathways targeted by SARS-CoV-2 to antagonize the innate immune response.

**Table 1 viruses-14-01247-t001:** Summary of mechanisms applied by SARS-CoV-2 to antagonize innate immune responses. The SARS-CoV-2 genome organization is represented, and proteins involved in host innate immunity pathway regulation are colored in red.

Mechanism of Inhibition		Viral Proteins	Refs.
**Evasion of sensing by host innate immune receptors** 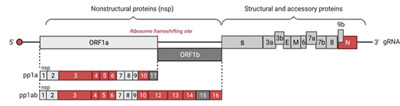	**Formation of DMVs**	**NSP3** **NSP4** **NSP6**	[30,31]
	**Capping**	**NSP10** **NSP12** **NSP13** **NSP14** **NSP16**	[32,33,34]
	**Blockage of RIG-I RNA recognition** *Interaction with its DExD/H helicase domain*	**N**	[35]
	**Cleavage of ISG15** *Antagonism of ISG15-dependent MDA5 activation*	**NSP3**	[36,37]
	**Cleavage of RIG-I at the last 10 N-terminal amino acids** *Blockage of its ability to signal through MAVS* *Promotion of the ubiquitination and proteosome-mediated degradation of MAVS*	**NSP5**	[38]
	**Inhibition of RIG-I CARD domain activation** *Interaction with Trim25*	N	[39,40,41]
**Inhibition of innate immune receptor signaling and IFNs production** 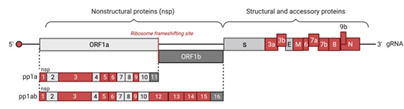	**Reduction of MAVS mediated IFN-β promoter activities**	**NSP1** **NSP3** **NSP5** **NSP12** **NSP13** **NSP14** **M** **N** **ORF3a** **ORF3b** **ORF6** **ORF7a** **ORF7b** **ORF8** **ORF9b**	[42,43,44,45,46]
	**Inhibition of IRF3 phosphorylation/nuclear translocation**	**NSP1** NSP5 NSP6 **NSP12** **NSP13** **NSP14** **NSP15** **ORF6** **ORF3b**	[42,43,44,49,50,51,62,63,64]
	**Cleavage of IRF3**	**NSP3** **NSP5**	[62,64]
	**Inhibition of MAVS signaling complex** *Interaction with TBK1* *Interaction with TOM70*	**NSP13** **M** **ORF7a** ORF9b	[19,44,54,55,56,57,58,59]
	**Inhibition of NF-κB** pathway *Interaction with Nup69 to block p65 translocation* *Blockage of Nemo K63-linked polyubiquitination* *Cleavage of TAB1*	NSP5 NSP9 ORF9b	[60,61,62]
**Inhibition of IFNs signaling and ISGs expression** 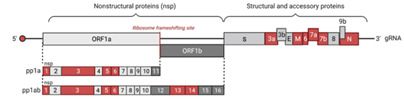	**Inhibition of STAT1/STAT2 phosphorylation**	**NSP1** **NSP6** **NSP13** **M** **N** **ORF3a** **ORF6** **ORF7a** **ORF7b**	[44,69,70]
	**Blockage of STAT1/STAT2 nuclear translocation** *Interaction with the nucleopore Nup98*	**ORF6**	[66,67,68]
	**IFNAR1 lysosomal degradation**	**NSP14**	[65]
	**Inhibition of ISGs**	**NSP3** **NSP5**	[36,43,76]
**Inhibition of protein production by targeting post-transcription and translation steps** 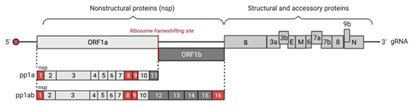	**Inhibition of pre-mRNA splicing** *Binding to U1 and U2*	**NSP16**	[71]
	**Disruption of protein trafficking** *Binding to SRP complex leading to the inhibition of signal peptide recognition*	**NSP8** **NSP9**	[71]
	**Blockage of mRNAs export** *Binding to the mRNA entry channel overlapping mRNA path* *Interaction with export receptor heterodimer NXF1-NXT*	**NSP1**	[71,72,73,74,75]

**Table 2 viruses-14-01247-t002:** Summary of discussed clinical trials with IFN therapies and their outcomes.

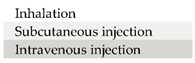				
Clinical Trial Name	Type of Trial	Type of Patients	Outcomes	Refs.
**IFN-α**				
**Nebulized IFN-α2b** with or without Arbidol	Uncontrolled, exploratory study	77 patients hospitalized with confirmed COVID-19 diagnosis (7 received IFN-α2b only, 46 IFN-α2b+ Arbidol)	Time to negative RT-qPCR significantly shorter in patients receiving inhaled IFN-α2b. Significant reduction in the duration of detectable virus in the upper respiratory tract. Reduced blood levels of inflammatory markers (IL-6, CRP).	[94]
**Nebulized IFN-α2b** with or without Umifenovir	Retrospective multicenter study	446 patients with confirmed COVID-19 diagnosis (242 received IFN-α2b, 216 early and 26 late)	Nebulized IFN-α2b initiation within 5 days of admission: - Associated with reduced mortality. - Not associated with hospital discharge or computed tomography (CT) scan improvement. Late initiation of nebulized IFN-α2b therapy: - Associated with delayed recovery. - Associated with increased mortality.	[98]
**Inhaled IFN-α2b**	Retrospective multicenter study	1401 patients with confirmed COVID-19 diagnosis (852 received IFN-α2b)	Early administration (3–5 days after symptom onset), associated with improved clinical outcomes: - Lower risk of disease progression. - Shorter hospitalization time. Delayed IFN-α2b intervention (6–8 days after symptom onset): associated with increased probabilities of risk events.	[99]
**PEG IFN-α2b** (subcutaneous injection) in addition to standard of care [antipyretics, cough suppressants, antibiotics, steroids, vitamins, anticoagulants, hydroxychloroquine and antivirals (e.g., Remdesivir)]	Multicenter, randomized, comparator-controlled, open-label phase 3 study	250 patients with moderate COVID-19 (120 received PEG IFN-α2b + standard of care)	Early viral clearance. Clinical status improvement. Decreased duration of supplemental oxygen.	[96]
**IFN-α2b** (subcutaneous injection) combined with Lopinavir/Ritonavir	Exploratory study	41 patients with confirmed COVID-19 diagnosis (19 received IFN-α2b + lopinavir/ritonavir)	Early administration of IFN-α2b within 72 h following admission- resulted in shorter hospital stay, (10 days compared with late administration - after 72 h following admission).	[97]
**IFN-β**				
**Nebulized IFN-β1a** (inhalation) **- SNG001** ClinicalTrials.gov Identifier: NCT04385095	Double-blind randomized, placebo-controlled, phase 2 pilot study	101 patients with confirmed COVID-19 diagnosis (50 received IFN-β1a SNG001)	Significantly greater odds of clinical improvement across the WHO Ordinal Scale for Clinical Improvement. Reduction of the odds of developing severe disease or dying.	[113]
**Nebulized IFN-β1b** (inhalation through a vibrating mesh aerogen nebulizer-Aerogen Solo) combined with Favipiravir ClinicalTrials.gov Identifier: NCT04385095	Randomized controlled open label study	89 adult patients hospitalized with moderate to severe COVID-19 (44 received IFN-β1b)	No significant differences in the inflammatory biomarkers at hospital discharge, in the overall lower hospital stay, transfers to the intensive care unit, neither in overall mortality.	[114]
**IFN-β1b** (subcutaneous injection) combined with Lopinavir, Ritonavir, Ribavirin	Multicenter, prospective, open label, randomized, phase 2 study	127 patients with mild to moderate COVID-19 (86 received IFN-β1b)	Administration within 7 days of symptom onset: - Suppression of the shedding of SARS-CoV-2. - Significant reductions in duration of RT-qPCR positivity and viral load (RT-qPCR negative by day 8). - Shorter time in complete alleviation of symptoms reduction in duration of hospital stay.	[107]
**IFN-β1a or IFN-β1b** (subcutaneous injections) combined with Lopinavir/Ritonavir COVIFERON trial ClinicalTrials.gov Identifier: NCT04343769	Randomized, open-label, controlled study	60 severely ill patients with positive RT-qPCR and Chest CT scans (20 patients assigned to IFN-β1a and 20 to IFN-β1b)	IFN-β1a: significant shorter time to clinical improvement. IFN-β1b: no significant difference. Lower numerically mortality both of the intervention groups (20% in the IFN-β1a group, 30% in the IFN-β1b group vs 45% in the control group) but not statically significant.	[108]
**IFN-β1a** (subcutaneous injection) in addition to the national protocol medications (Hydroxychloroquine plus Lopinavir- Ritonavir or Atazanavir-Ritonavir) Clinical Identifier: IRCT20100228003449N28	Randomized, open-label, controlled study	92 patients with severe COVID-19 (42 received IFN-β1a)	No change the time to reach the clinical response. Length of intensive care unit and hospital stays and duration of mechanical ventilation not statistically different. Significantly increased discharge rate on day 14. Early administration significantly reduced mortality.	[109]
**IFN-β1a** (subcutaneous injection) combined with Remdesivir ClinicalTrials.gov Identifier: NCT04492475	Randomized, double-blind, placebo-controlled study	969 patients hospitalized COVID-19 patients with presence of radiographic infiltrates on imaging, a peripheral oxygen saturation on room air of 94% or less, or requiring supplemental oxygen (487 received IFN-β1a)	No clinical improvement. Worse outcomes after treatment IFN-β1a in patients who required high-flow oxygen at baseline.	[111]
**IFN-β1a** (subcutaneous injection). For patients receiving high-flow oxygen, ventilation, or extra- corporeal membrane oxygenation: intravenous interferon. ClinicalTrials.gov Identifier: NCT04315948	Randomized, double-blind, placebo-controlled study	4127 (2063 received IFN-β1a)	No effect on hospitalized patients (based on overall mortality, initiation of ventilation, and duration of hospital stay).	[110]
**IFN** **-** **λ**				
**PEG IFN-****λ** (subcutaneous injection) ClinicalTrials.gov Identifier: NCT04354259	Randomized, double-blind, placebo-controlled study	60 outpatients with COVID-19 (30 received PEG IFN-λ)	Greater decline in RT-qPCR with viral clearance by day 7. Prevent clinical deterioration and shorten duration of viral shedding.	[128]
**PEG IFN-****λ** (subcutaneous injection) ClinicalTrials.gov Identifier: NCT04331899	Randomized, double-blind, placebo-controlled phase 2 study	120 outpatients with mild to moderate COVID-19 (30 received PEG IFN-λ)	No shortened duration of SARS-CoV-2 viral shedding. No improved of symptoms.	[129]

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
