# Peer review of "Characterization of SARS-CoV-2 Evasion: Interferon Pathway and Therapeutic Options"

_viruses, 2022, doi:10.3390/v14061247_

Round 1

Reviewer 1 Report

Review report on manuscript 1724540 submitted to Viruses

Characterization of SARS-CoV-2 evasion : interferon pathway and therapeutic options

M Znaidia, C Demeret, S van der Werff, AV Komarova

Znaidia et al present in this review what is currently known about the interplay between SARS-CoV-2 proteins and innate immune response pathway, i.e. the interferon and ISGs activation pathways. They are describing the evasion of the innate immune response system by SARS-CoV-2 very clearly and step-by-step. Then, they are summarizing the different results concerning in vitro, in vivo and clinical trials of interferon-based treatments on COVID-19 patients so far published.

The manuscript is very interesting and quite comprehensive. Nevertheless, the authors should include how viral mRNA are escaping the translation inhibition mediated by NSP1 when interacting with the 40S ribosome (host shutoff role of NSP1) (Mendez et al. Cell Reports, 2021).

Some minor points :

Page 6 : the 7SL RNA is not a specific region of SRP but a non-coding RNA that is a constituent of the SRP (which is a ribonucleoprotein complex composed of 1 RNA (7SL) and 6 proteins).

Page 7 : up today => up to now

Page 8 (table 2, 3d line) : morality => mortality

Page 10 : IFN- => IFN-λ

Double-check the references, some are not complete (for instance ref #74), some have missing characters (for instance ref #47)

Author Response

We thank Reviewer #1 for positive comments on our review manuscript. We have accounted for the comments by Reviewer #1 in the revised manuscript.

Mendez et al. Cell Reports, 2021 study that suggests the mechanism of how viral mRNA are escaping the translation inhibition mediated by NSP1 when interacting with the 40S ribosome has been discussed (lines: 231-236).

Page 6: the 7SL RNA is not a specific region of SRP but a non-coding RNA that is a constituent of the SRP (which is a ribonucleoprotein complex composed of 1 RNA (7SL) and 6 proteins).

Has been corrected.

Page 7: up today => up to now

Has been corrected.

Page 8 (table 2, 3d line): morality => mortality

Has been corrected.

Page 10: IFN- => IFN-λ

Has been corrected.

Double-check the references, some are not complete (for instance ref #74), some have missing characters (for instance ref #47)

Has been verified.

Reviewer 2 Report

This is a very well-written, informative, and comprehensive review. The reviewer has a few minor comments as follows:

1) IFN should be expanded at first use.

2) Few symbols are misspelled: NF-κB and IKKß (line 64, 160, 162, 163, 168)

3) line 52: RIG-I-like

4) line 197, while describing ORF7A mediated inhibition of ISGF3 translocation, the authors can expand on the ubiquitination of ORF7A itself that is important for its function.

5) line 243-244, 293: The authors could point out that pre-treatment of IFNa and IFNb inhibits virus infectivity/production but not post-treatment ( Chen et al. Cell Death Discovery 2021;7:114; Bessière P, et al PLoS Pathog 17(8): e1009427.; Lokugamage et al. J. Virol. 2020, 94, e01410-20 etc.) that is relevant for describing the timing of administration of IFN treatment (line 262- 273).

6) The authors could describe more the potential for inhibitors targeting the virus protein and host protein interactions and inhibitors of viral proteases like GRL0617 (Fu et al. Nature Communications 2021; Shin et al. Nature 2020 etc.)

Author Response

We thank Reviewer #2 for positive comments on our review manuscript. We have accounted for the comments by Reviewer #2 in the revised manuscript.

1) IFN should be expanded at first use.

Has been expanded (lines 61-62).

2) Few symbols are misspelled: NF-κB and IKKß (line 64, 160, 162, 163, 168)

Has been corrected.

3) line 52: RIG-I-like

Has been corrected.

4) line 197, while describing ORF7A mediated inhibition of ISGF3 translocation, the authors can expand on the ubiquitination of ORF7A itself that is important for its function.

Has been expanded (lines 205-207).

5) line 243-244, 293: The authors could point out that pre-treatment of IFNa and IFNb inhibits virus infectivity/production but not post-treatment ( Chen et al. Cell Death Discovery 2021;7:114; Bessière P, et al PLoS Pathog 17(8): e1009427.; Lokugamage et al. J. Virol. 2020, 94, e01410-20 etc.) that is relevant for describing the timing of administration of IFN treatment (line 262- 273).

Has been added on lines 400-402.

6) The authors could describe more the potential for inhibitors targeting the virus protein and host protein interactions and inhibitors of viral proteases like GRL0617 (Fu et al. Nature Communications 2021; Shin et al. Nature 2020 etc.)

Has been described (lines 443-446).